# Preparation, Characterization and Application of a Low Water-Sensitive *Artemisia sphaerocephala* Krasch. Gum Intelligent Film Incorporated with Anionic Cellulose Nanofiber as a Reinforcing Component

**DOI:** 10.3390/polym12010247

**Published:** 2020-01-20

**Authors:** Tieqiang Liang, Lijuan Wang

**Affiliations:** Key Laboratory of Bio-based Materials Science and Technology of Ministry of Education, Northeast Forestry University, Harbin 150040, China; tie_qiang@163.com

**Keywords:** *Artemisia sphaerocephala* Krasch. gum, anionic cellulose nanofiber, red cabbage extracts, pH-sensing film, low water-sensitivity

## Abstract

A low-water-sensitive *Artemisia sphaerocephala* Krasch. gum (ASKG) based intelligent film was developed. Red cabbage extracts (RCE) was selected as a natural pH-sensitive indicator, and anionic cellulose nanofiber (ACNF) was added as a hydrophobic and locking host. The zeta potential, rheology, Fourier-transform infrared spectroscopy, X-ray diffractometry, and release results indicated that the RCE was locked by the ACNF via electrostatic interactions, moreover, broke the original complicated network and ordered arrangement of polymer molecules in the developed intelligent films. RCE addition decreased the tensile strength, oxygen, and water vapor barrier properties and light transmission of the developed intelligent films, while increasing the elongation at break. The films could respond to buffer solutions and NH_3_ through different color changes. The developed intelligent film was hydrophobic, which could precisely detect the freshwater shrimp freshness in real time via color changes, which indicated that the films have potential in intelligent packaging and gas-sensing label fields.

## 1. Introduction

Biomass packaging materials have been extensively investigated to replace synthetic plastics, because of their low cost, easy processability, non-toxicity, and biodegradability, as awareness of environmental protection and food safety has increased [1]. In recent years, commercial intelligent labels or films, such as the Timestrip^®^, 3M™MonitorMark™, CheckPoint^®^, and Ageless Eye^®^, appear in our daily lives [2]. However, the raw materials and indicators that are used are organic synthetic substances or heavy metals, which can harm the ecosystem and people’s health, and pollute packaged food. Naturally degradable and edible intelligent films or labels have received extensive attention from researchers to resolve this problem. In recent years, polysaccharides, such as gelatin [3], chitosan [4], gellan gum [5], and starch [6] based intelligent films, have been investigated to detect seafood and meat freshness and they showed good detecting effects.

Synthetic indicators, such as methylene blue [7], bromocresol green [8], and chlorophenol red [9], were used first in intelligent packaging films. However, these indicators can pollute packaged food and may pose a harmful risk to the ecosystem and people’s health [10]. To resolve those issues, natural indicators, such as curcumin [11,12,13], anthocyanins from black carrot [14], red cabbage [15,16], purple-fleshed sweet potato [17], and roselle [18], have been used to replace synthetic indicators in edible intelligent packaging films according to recent reports [19], because they are non-toxic [20] and can change color in different pH conditions [21]. Among them, red cabbage has been extensively used as a raw material because of its wide range of sources and low cost. Red cabbage extracts (RCE) operate over a wide spectrum at different pHs [22]. 

In our previous work, a series of intelligent films was developed, in which *Artemisia sphaerocephala* Krasch. gum (ASKG) was selected as a substrate [23,24,25]. The ASKG was a suitable film-forming material for intelligent food packaging films, however, the developed intelligent films showed a high water-sensitivity, because of the high hydrophilic property of all film-forming materials and extracts, which limits their practical application. This defect also exists in other similar films [26,27,28].

To resolve this defect, some scholars have selected nano-crystalline cellulose (NCC), cellulose nanofiber (CNF), or their derivatives as a hydrophobic material to reduce the film’s water sensitivity, because they are green, sustainable, and eco-friendly [28,29,30,31,32]. Among them, anionic cellulose nanofiber (ACNF) is a nanosized flexible fibril (5–50 nm width and micron-scale length) with negative charge, which could disperse evenly in water [33,34]. ACNF has been extensively investigated, because of its high crystallinity, dispersibility, desirable environmental friendliness, and reactivity [34,35,36]. No report exists regarding the use of ACNF in ASKG-based films.

In this work, RCE was extracted as a natural indicator and ACNF was used as a locking and reducing water-sensitive compound. A high-sensitive intelligent film with a low water sensitivity based on ASKG was prepared. The rheology and zeta potential of every film-forming solution was determined to investigate its film-forming property and the interactions between the ACNF and RCE. The structural, morphological, mechanical, barrier, optical, thermal, and hydrophobic properties of the developed films were also characterized. The locking effects between the ACNF and RCE were evaluated through release test, the chroma-response of the developed intelligent films in solutions with different pH of 3.0–10.0, and a NH_3_ atmosphere under different humidities (33%, 75%, and 90%) was also observed. An industrial production was developed and used to detect the freshwater shrimp freshness in real time, and the relationship of the response characteristics between them was also constructed to assess the practical application of the developed intelligent film.

## 2. Materials and Methods

### 2.1. Materials and Regents

ASKG was obtained from Zhengzhou Yuhe Food Additive Co., Ltd. (Zhengzhou, China). ACNF (width of 35 nm, length >1 μm, viscosity of 355 MPa·s) was obtained from Chemkey Advanced Materials Technology Co., Ltd. (Shanghai, China). Edible sodium paraben was obtained from Jingrun Bio-Tech. Co., Ltd. (Shenzhen, China). Analytical glycerol was obtained from Kemiou Chemical Reagent Co., Ltd. (Tianjin, China). Red cabbage was obtained from Songlei Supermarket (Harbin, China). Buffer solutions (pH 3.0–10.0) were obtained from Beijing Wanjia Shouhua Biotechnology Co., Ltd. (Beijing, China).

### 2.2. Experimental Methods

#### 2.2.1. Extraction and Quantification of RCE

RCE was extracted, according to the literature [25], 50 g of red cabbage powder and 1 L of 50% (*v*/*v*) ethanol solution were mixed and then treated under ultrasound of 540 W for 1 h at 50 °C, then, filtered by using Buchner funnel to obtain the RCE solution, in which the anthocyanin content was 0.4717 mg/mL, according to the literature through a pH-differential method [37], and the solid content of RCE solution was 43.77 g/L through a constant-mass method.

#### 2.2.2. Preparation of ASKG/ACNF/RCE Intelligent Films

The preparation method from our previous works was used with some modifications [23,24,25]. The optimal mass ratio of ASKG and ACNF was 19:1 (total mass was 6 g), according to preliminary experiments. Based on the optimal mass ratio, 6 g of ASKG and ACNF was dispersed in 600 mL of distilled water and then heated for 30 min at 60 °C with constant stirring at 800 rpm. Subsequently, 40% (weight ratio, ACNF and ASKG basis) glycerol and 0–15% (weight ratio, ACNF, ASKG and glycerol basis) RCE was mixed in and then stirred for a further 20 min. The developed films were obtained through casting method by using an acrylic resin plate (26 cm × 26 cm × 4 cm) after 50 h at 60 °C in a drying box and recorded as AFR0, AFR5, AFR10, and AFR15. 

#### 2.2.3. Preparation of Intelligent Films to Detect Freshwater Shrimp Freshness

The intelligent film preparation (AFR-10P) to detect the freshwater shrimp freshness was as follows.

According to Section 2.2.2, 6 g of ASKG and ACNF were dispersed in distilled water and then heated for 30 min at 60 °C with constant stirring at 800 rpm. Subsequently, 40% glycerol and 7.0% RCE were added and stirred for a further 20 min. Thereafter, 0.05% sodium paraben (a food preservative) was added and stirred for a further 20 min according to Chinese National Standard GB 2760. The developed films were obtained through casting after 60 h at 50 °C in a drying box and recorded as AFR-10P.

### 2.3. Characterization

#### 2.3.1. Zeta Potential Changes of Solutions

According to our previous report, 1 mL of samples was added to a specific cell and then tested at 25 °C on a Zetasizer Nano ZS (under He–Ne laser at 90° scattering angle, Malvern, Worcestershire, UK) [25].

#### 2.3.2. Rheological Analysis of Solutions

The rheology test was conducted according to the literature [25]. The shear rate ranged from 0.1 to 100 s^−1^ and the test temperature was 25 °C.

The flow behavior was assessed by using a Cross model with the following equation:*η* = *η_∞_* + (*η*_0_ − *η_∞_*)/[1 + (*λ_c_·γ*)*^n^*](1)
where *η*, *η*_0_, *η_∞_*, *λ_c_*, *γ*, and *n* are the surface viscosity (Pa·s), viscosity at zero shear rate (Pa·s), infinite viscosity (Pa·s), a time constant, shear rate (s^−1^), and a dimensionless exponent, respectively.

The dynamic rheological properties were described by data of storage modulus (G′, Pa), loss modulus (G″, Pa), and compound viscosity (*η**, Pa·s) from 0.1 to 100 rad/s under 0.3% of strain.

#### 2.3.3. Fourier-Transform Infrared Spectroscopy

The Fourier-transform infrared (FT-IR) spectra of the RCE and ASKG/ACNF/RCE intelligent films were performed on a Thermo Fisher Scientific spectrometer (ATR model, Waltham, MA, USA) over a wavenumber range of 4000–600 cm^−1^ at a resolution of 4 cm^−1^.

#### 2.3.4. X-ray Diffractometry Spectroscopy

The X-ray diffractometry (XRD) patterns of the ASKG, ACNF, and ASKG/ACNF/RCE intelligent films were obtained on a diffractometer (D/max-2200, Rigaku, Tokyo, Japan) with a scanning mode by using Cu–Kα radiation from 5–40° with 2°/min of scanning rate.

#### 2.3.5. Thermogravimetric Analysis

The thermostability of the ASKG/ACNF/RCE intelligent films was compared through thermogravimetric analysis (TGA) and derivative thermogravimetric analysis (DTG) on a TGA Q500 TA instrument (Thermo Fisher Scientific, Waltham, MA, USA) from room temperature to 600 °C at 10 °C/min under 99.999% of nitrogen atmosphere.

#### 2.3.6. Morphology Analysis

The micro-morphology of the freeze-dried ACNF was tested on a transmission electron microscope (JEM-2100, JEOL Co., Ltd., Tokyo, Japan) at an accelerating voltage at 200 kV and a 5000× magnification. After spraying a thin layer of gold, the fractured section and surface morphology of the samples were tested by using SIGMA 500 high-resolution field emission scanning electron microscope (Carl Zeiss, Jena, Germany), at 6 kV of acceleration voltage.

#### 2.3.7. Mechanical Properties

All films were stored at a 43% relative humidity for 48 h before testing to eliminate the influence of moisture on the test results. The thickness of the samples (strip of 15 mm × 80 mm) was tested by using ID-C112XBS micrometer (Mitutoyo Corp., Tokyo, Japan) and tensile strength (TS), and elongation at break (EB) of the samples (strip of 15 mm × 80 mm) was tested by using XLW-PC magnum experiment machine (Labthink, Jinan, China) at a strain rate of 300 mm/min.

#### 2.3.8. Barrier Properties

The film oxygen permeability (OP) was tested by using OX2/230 oxygen permeability tester (Labthink, Jinan, China), according to the instructions.

The film water vapor permeability (WVP) was tested according to the literature [25]. A container that contained absolutely anhydrous calcium chloride (CaCl_2_) was sealed with the samples and weighed every 30 min at 25 °C and a 75% relative humidity. When the mass stopped changing, the WVP values were calculated according to related parameters.

#### 2.3.9. Light Transmittance Analysis

The light transmittance of the samples was scanned on an UV-2600 ultraviolet (UV)–visible (Vis) spectrophotometer (Shimadzu, Tokyo, Japan) over all th wavelengths at 25 °C.

#### 2.3.10. Release Test

0.05 g of AFR15 film was added to 10 mL of different concentrations of ethanol solution (75% and 100%, *v*/*v*) and oscillated for 12 h at 100 rpm at room temperature. The residue solution spectra were scanned from 475–800 nm on UV-2600 UV–Vis spectrophotometer.

#### 2.3.11. Water-Sensitivity Test of Intelligent Films

The water sensitivity of the intelligent films was evaluated from the water-contact angle, which was determined on an OCA20 video optical-contact-angle measurement instrument (Dataphysics, Filderstadt, Germany) by the pendant-drop method at 25 °C. Deionized water (5 μL) was automatically pressed with an injector onto the intelligent film surface, and the contact angle data of each intelligent film were analyzed and collected by a CCD video system on the computer.

#### 2.3.12. Chroma-Response Test

The *L**, *a**, *b**, and Δ*E* values of the samples (strips: 20 mm × 40 mm) at different pHs and concentrated NH_3_ condition were tested by using CM-2600d colorimeter (KONICA MINOLTA, Tokyo, Japan).

The chroma-response of the intelligent films to buffer solutions (pH = 3.0 to 10.0) was tested after the samples were immersed in 10 mL of buffer solutions for a certain time. Subsequently, the sample’s color was tested.

Before NH_3_-response test, the samples were stored first in containers with relative humidity of 33%, 75%, and 90% for 12 h at 25 °C. Afterwards, 5 mL of ammonia was added into the containers. The chroma parameters of the samples were tested after 10 min.

#### 2.3.13. pH and Total Volatile Basic Nitrogen of Freshwater Shrimp

The freshwater shrimp pH was tested according to GB 5009.237-2016 (a Chinese National Standard). Briefly, 10 g of sample was added to 100 g of 0.1 mol/L KCl solution and then homogenized for 30 min. Subsequently, the mixture pH was tested by using PHSJ-3F pH meter (INESA, Shanghai, China).

The total volatile basic nitrogen (TVB-N) of freshwater shrimp was tested according to Chinese National Standard GB 5009.228-2016. Briefly, 20 g of milled sample was added to 100 mL of distilled water, which was cooled after boiling and oscillated for 30 min. Next, the mixture was centrifuged at 3000 rpm for 10 min. Afterwards, 5 mL of supernatant was alkalized by 5 mL of MgO suspension (10 g/L). Steam distillation was conducted on a semi-micro Kjeldahl analyzer (Beijing Bonokin Technology Co. Ltd., Beijing, China) for 5 min and then absorbed by 10 mL of 20 g/L boric acid solution. Finally, the boric acid solution containing TVB-N was precisely titrated with 0.01 mol/L HCl solution. The TVB-N value was calculated and expressed in mg/100 g.

#### 2.3.14. Freshness Detection Test

Before testing, the color parameter of the AFR-10P film (15 mm × 15 mm, ~0.01 g) was tested. The sample was placed at a 43% humidity for 24 h. Subsequently, 25 g of freshwater shrimp was used in the experiment at 20 °C. When the color of the film changed, the color parameter should be tested immediately. Figure 1 shows the specific test diagram.

#### 2.3.15. Statistical Treatment

SPSS 19.0 software was used to analyze all the data and Duncan multiple comparisons (*p* < 0.05) was used to analyze the discrepancies among them. The results are expressed in lowercase letters in figures and superscripts in tables.

## 3. Results and Discussions

### 3.1. Zeta Potential Analysis

The related data of RCE, ACNF, AFR0, AFR5, AFR10, and AFR15 solutions in Figure 2 indicate that the charge of the RCE solution is −6.1 mV [38] and the charge of ACNF solution is −47.0 mV [39]. The charge of the AFR0 solution is −17.4 mV, which is related to the offset between the positive and negative charges [39]. When RCE increased from 0% to 15%, it also increased from −17.4 to −12.3 mV, which indicates that an electrostatic interaction between the flavylium cations in RCE and the COO^−^ in ACNF occurred.

### 3.2. Rheological Analysis

#### 3.2.1. Steady Rheological Analysis

Figure 3 shows that the viscosity of all film-forming solutions decreased with the shear rate increased (characteristics of non-Newtonian fluids), which indicates that the hydrogen bonds that were formed by ASKG, ACNF, glycerol, and RCE were destroyed by the shearing force, and the original complicated network structure could not be restored within a certain time [40]. This result is attributed to the small molecular structure of RCE that could penetrate the complicated network structure easily formed among each content, and new hydrogen bonds formed between the ASKG, ACNF, glycerol and RCE. The complex that formed between the RCE and ACNF by the electrostatic attraction also broke the complicated network structure in film-forming solutions.

Table 1 shows that the Cross model is suitable for fitting the ASKG/ACNF/RCE film-forming solutions (*R*^2^ > 0.999), and the *η*_0_ and *K* values decreased from 1.3857 to 1.3015 Pa·s and from 2.1880 to 1.8871 s with an increase of RCE from 0% to 15%, respectively. These results are related to the RCE addition, which broke the complicated network structure that formed between the ASKG, ACNF, and glycerol. Entanglements among the ACNF chains weakly decreased the weakly. Moreover, *p* values ˂ 1 show that all of the solutions were pseudoplastic fluids [41]. These results reveal that RCE changed the network formed by ASKG, ACNF and glycerol molecules.

#### 3.2.2. Dynamic Rheological Analysis

As shown in Figure 4, all of the film-forming solutions were weak gel systems, because a crossover point between G″ and G′ curves existed [42]. The *η** value decreased as angular frequency increasing. The crossover point between G′ and G″ shifted from 3.09 to 3.58 rad/s with an increase of RCE from 0% to 15%, which was attributed to the electrostatic attraction between RCE and ACNF changed the complicated network structure. These results show that RCE exhibited plasticizing effects, to some extent. However, RCE addition did not change the network system properties of the film-forming solutions [43]. These results are agreed with the steady rheological analysis.

### 3.3. FT-IR Analysis

The RCE spectrum shows that the bands at 3308, 1638, 1414, and 1045 cm^−1^ were related to the O–H stretching vibration in hydroxyl, the C=C stretching vibration in aromatic ring, the C–O bending vibration in phenols, and the O–C stretching vibration in flavonoid, respectively [22]. Figure 5b shows that the bands at 3289, 2927, and 2885 cm^−1^ of AFR0 film were related to the O–H stretching vibration in hydroxyl, the C–H stretching vibration of the film components, respectively. Bands at 1644 and 1416 cm^−1^ were related to the C=O stretching vibration and –COO– stretching vibration [44,45]. The bands at 1021, 920, and 868 cm^−1^ were related to the characteristic stretching vibration of C–O and O–C in pyran ring, respectively [46].

After RCE addition, the band at 3289 cm^−1^ shifted to a low wavenumber and the band at 2924 cm^−1^ was enhanced, which indicates that the hydrogen bonds among the ASKG, ACNF and glycerol were broken and new hydrogen bonds formed [47]. Bands at 1644 and 1412 cm^−1^ broadened and shifted to a low wavenumber, which indicates that an electrostatic attraction occurred between RCE and ACNF [22]. All of those results show that RCE was locked into the intelligent film through electrostatic attraction with ACNF.

### 3.4. XRD Analysis

Figure 6a shows that a weak peak at 11.8° and two broad peaks at 16.4° and 22.2° were the characteristic polysaccharide peaks of ASKG. Figure 6b shows two peaks at 15.4° and 22.5° in the XRD pattern of ACNF, which indicates a cellulose I crystal structure [48].

The XRD pattern of the AFR0 film (Figure 6c) shows three characteristic peaks at 10.9°, 16.3°, and 22.3°. The diffraction peaks indicate that ACNF rearranged ASKG molecular chains and the chains blended. After RCE addition, the peak around 16.3° was enhanced, which indicates that RCE addition broke the complicated network structure that formed between ASKG and ACNF by electrostatic interaction. The peak at 16.3° decreased with a further increase in RCE, which indicates that the electrostatic interaction between RCE and ACNF changed the crystal structure of the intelligent film.

### 3.5. Thermogravimetric Analysis

Figure 7 shows that three mass loss peaks can be observed in AFR0 film; the mass loss peaks at 69.07, 186.04, and 287.81 °C were the loss of adsorbed water [49], the decomposition of glycerol [23], and the decomposition of ASKG and ACNF [23,50]. After RCE addition, the thermal decomposition temperature of the third mass loss peak shifted from 287.81 °C to 281.38 °C, which indicates that the RCE broke the tightly complicated network structure between the ASKG and ACNF chains. 

### 3.6. TEM and SEM Observation

Figure 8 exhibits the ACNF morphology and the flat/cross-sectional surfaces of the intelligent films. Figure 8A shows that the 1% of ACNF solution was a transparent milky white, which indicates that the ACNF was at the nanoscale. The ACNF showed a filament structure with a nanometer diameter and a micron length, according to the TEM image. The filaments were entangled.

As shown in Figure 8B, all of the flat surfaces of the intelligent films were homogeneous, but wrinkled, which is attributed to the long filament morphology of ACNF. Layered structures were visible in the cross-section surface of the AFR0 film, because the peeling effects between the ACNF and ASKG during brittle fracture in liquid nitrogen. After RCE addition, the layered structures in the cross-sectional surfaces of the intelligent films increased, which indicates that the electrostatic attraction between RCE and ACNF broke the arrangement of molecular chains and made the peeling effects of ACNF easier. A further increase in the amount of RCE added, the layered structures in the cross-section surfaces decreased, because new hydrogen bonds formed among the film-forming polymers and the rearrangement of molecular chains occurred. Based on all of the above analyses, the schematic structure of the ASKG/ACNF/RCE intelligent film is shown in Figure 9.

### 3.7. Mechanical Properties

Figure 10 shows that the thickness of the intelligent films slightly increased from 0.061 mm to 0.066 mm, owing to the increase of solid polymers, moreover, the tensile strength decreased from 43.23 to 24.83 MPa, but the elongation at break increased from 56.13% to 75.87% with the increase of RCE from 0% to 15%.

### 3.8. Oxygen and Water Vapor Permeabilities

Figure 11 shows that the OP and WVP values of AFR5, AFR10, and AFR15 intelligent films were higher than the AFR0 intelligent film, owing to the electrostatic attraction between RCE and ACNF that broke the tight structure that formed by ASKG, ACNF, and glycerol. With an increase in RCE from 10% to 15%, the two values slightly decreased from 0.0169 to 0.0115 ((cm^3^·mm)/(m^2^·day·atm)) and from 4.5859 to 3.8494 × 10^−10^ g/(s∙m∙Pa), respectively. This result is attributed to the formation of a new tight network structure that formed by RCE, ASKG, ACNF and glycerol [51]. The developed intelligent films retained excellent oxygen and water vapor barrier properties.

### 3.9. Light Transmission of Intelligent Films

Figure 12 shows that the light transmission of the intelligent films decreased from 49.96% to 36.75% with an increase in RCE from 0% to 15% (at 600 nm), owing to the RCE, broke the ordered arrangement of ASKG and ACNF molecular chains, so that the light scattering and refraction increased. The entangled structure of the ACNF also increased the light scattering and reflection. The intelligent films showed UV light shielding effects when the amount of RCE added exceeded 5%, which indicates that the AFR10 and AFR15 films could reduce food spoilage that is caused by UV light.

### 3.10. Water-Sensitivity Analysis

Figure 13 shows the water-contact-angle data of the ASKG/ACNF/RCE and ASKG/CMC·Na/RCE intelligent films. It shows that the contact angle of the AFR0 film was 102°, which indicates a low water sensitivity, owing to the excessively tight network structure of the AFR0 film and the hydrophobic properties of the ACNF. An increase of RCE resulted in a decrease of contact angle. However, the contact angle remained at 93.25° when the amount of RCE added reached 15%, which indicates a low water sensitivity. These results were attributed to the RCE addition that broke the tight structure that formed among the ASKG, ACNF and glycerol, and promoted water penetration. When compared with the ASKG/CMC·Na/RCE intelligent films in our previous work [25], the contact angle of films that contained ACNF were higher than those that contained CMC·Na, which made them suitable for use under higher humidity conditions.

### 3.11. Release Analysis

The release test that was based on the AFR15 film was conducted for evaluating the locking effects between the ACNF and RCE. Figure 14 shows that a maximum absorbance of the RCE solution could be observed at 538 nm, with a value of 0.089. The maximum absorbance of the 75% and 100% ethanol filtrates were 0 and the two filtrates were colorless after intelligent film immersion and oscillation for 12 h, which indicates that the RCE was locked in the ACF15 film. The same result was obtained for the AFR5 and AFR10 films. 

### 3.12. Buffer Solutions and NH_3_ Atmosphere Response Analysis of Intelligent Films

#### 3.12.1. Chroma-Response in Buffer Solution

The intelligent film was dark-reddish-purple, grayish-purple, dark-brown, atropurpureus, and yellowish-green at a pH of 3.0, 4.0–6.0, 7.0, 8.0–9.0, and 10.0 in Table 2, respectively. With an increase in pH from 3.0 to 6.0, the *a** value decreased, but the *b** value showed the opposite trend, owing to an increase in the pseudo-base carbinol structure of the RCE. When the pH increased to 7.0, the *a** and *b** values decreased further, owing to the change in RCE structure from pseudo-base carbinol to the anionic form. A further increase in pH to 10.0 resulted in a decrease of the *a** value to negative and an increase in the *b** value, because the change in RCE structure to chalcone [52]. The Δ*E* values of the AFR10 and AFR15 films exceeded 5, which indicates that the color differences could be observed by the eye [53].

#### 3.12.2. Chroma-Response in NH_3_ Condition

NH_3_ was selected to simulate the TVB-N that is produced by seafood to evaluate the practical application of the ASKG/ACNF/RCE intelligent films. As shown in Table 3, the *L** values decreased as the RCE addition and relative humidity increased, which indicates that the developed film became dark. After increasing the relative humidity to 75%, the *a** and the *b** values decreased, owing to the acceleration of the contact between RCE and OH^−^ ions by the hydrophobic ACNF, which accelerated the structural change in RCE.

When compared with ASKG/CMC·Na/RCE intelligent films in our previous work [25], the color change was slightly different, owing to the scattering and reflection of light that was caused by the long fiber morphology of the ACNF.

### 3.13. Detecting Analysis of Intelligent Film for Freshwater Shrimp Freshness in Real Time

Figure 15 shows changes in pH and TVB-N of the freshness shrimp, colorimetric parameters, and a photograph of the intelligent film and they are listed in Table 4. The pH and TVB-N of the fresh freshwater shrimp were 6.831 and 1.4047 mg/100 g, respectively. The intelligent film was dark purple. After 12 h, the two values increased to 7.163 and 20.0759 mg/100 g, respectively. Table 4 shows that the *a** value decreased to negative, but the *b** value showed opposite changes. The intelligent film was yellow–green. The pH of the freshwater shrimp changed to alkaline and spoilage occurred after 12 h according to GB 2733-2015 (a Chinese National Standard). A further increase in storage time showed a more obvious increase in the two values. These results are attributed to protein and amino acid decomposition in freshwater shrimp into volatile nitrogen under the action of microorganisms [54]. According to our study, ~0.01 g of intelligent film was suitable for detecting freshness information for 25 g of freshwater shrimp to consumers through a color change in real time.

## 4. Conclusions

An intelligent and low water-sensitive ASKG-based film was developed and ACNF was selected as a reinforcing and locking compound to reduce films’ water sensitivity and lock RCE. All of the film-forming solutions showed the characteristics of non-Newtonian fluids. The zeta potential and rheological results of the film-forming solutions indicated that RCE interacted with ACNF through electrostatic attraction and changed the complicated network structure in the film-forming solutions. The FTIR results were consistent with the above results. The XRD results showed that RCE addition changed film’s crystalline state. The TG results revealed that RCE did not obviously change the stability of the intelligent films. TEM observation showed that ACNF was a filament structure with a nanometer diameter and a micron length. SEM observation showed that film’s flat surfaces were homogeneous, but wrinkled, and the film’s cross-section showed a layered structure. RCE addition reduced the TS and transmittance of the intelligent films from 43.23 to 24.83 MPa and 49.96% to 36.75% at 600 nm, but it increased the EB from 56.13% to 75.87%. The OP and WVP of films that contained RCE was higher that the film without RCE, but the OP and WVP decreased from 0.0169 to 0.0115 ((cm^3^·mm)/(m^2^·day·atm)) and from 4.5859 × 10^−10^ to 3.8494 × 10^−10^ g/(s∙m∙Pa) with an increase in RCE addition from 10% to 15%. The release results indicated that RCE has been locked in the intelligent films. The intelligent films could respond to buffer solutions and NH_3_ atmosphere through color changes. The contact angles of the intelligent films that contained ACNF were higher than those with CMC·Na. Moreover, 0.01 g of the AFR-10P film could detect the freshness information for 25 g of freshwater shrimp in real time and its color changed from dark-purple to yellow–green when the samples started to spoil. The developed films show potentials in edible and intelligent food packaging field.

## Figures and Tables

**Figure 1 polymers-12-00247-f001:**
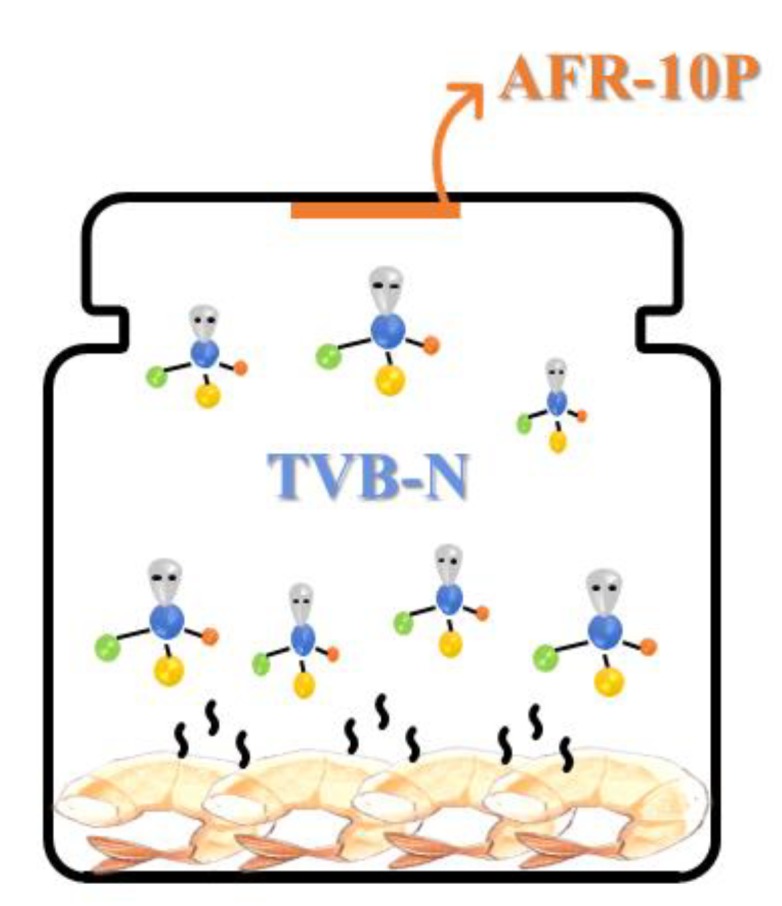
Detecting experiment schematic.

**Figure 2 polymers-12-00247-f002:**
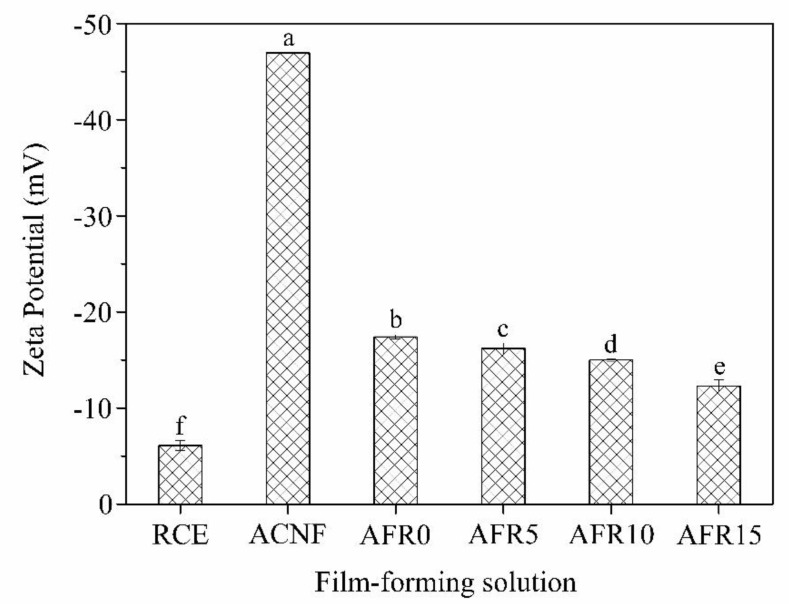
Zeta potential of film-forming solution with different red cabbage extracts (RCE) amounts.

**Figure 3 polymers-12-00247-f003:**
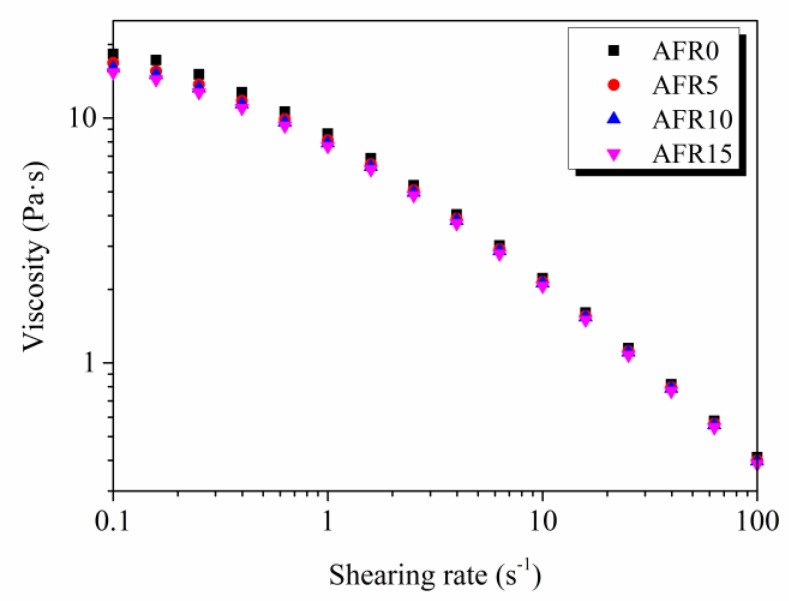
Effect of the RCE amount on steady rheological properties of film-forming solutions.

**Figure 4 polymers-12-00247-f004:**
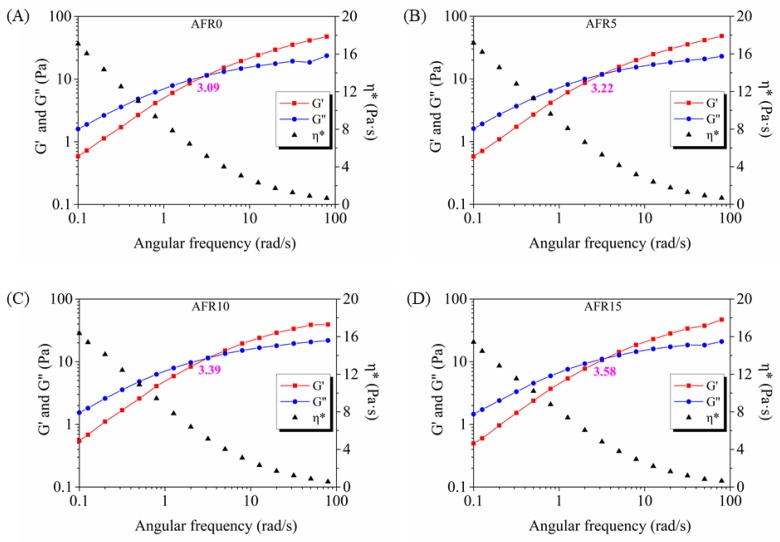
Effect of the RCE amount on dynamic rheological properties of film-forming solutions: AFR0 film-forming solution (**A**), AFR5 film-forming solution (**B**), AFR10 film-forming solution (**C**), and AFR15 film-forming solution (**D**).

**Figure 5 polymers-12-00247-f005:**
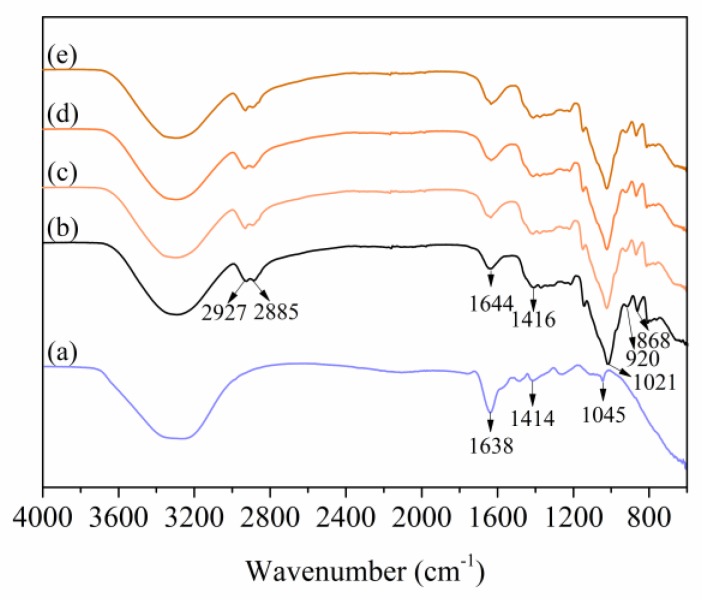
Fourier-transform infrared (FT-IR) spectra of RCE and intelligent films: RCE (**a**), AFR0 (**b**), AFR5 (**c**), AFR10 (**d**), and AFR15 (**e**).

**Figure 6 polymers-12-00247-f006:**
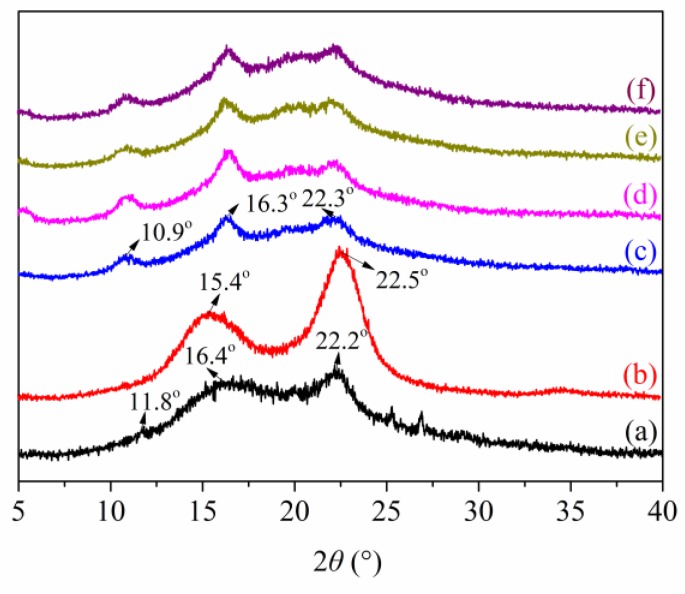
X-ray diffractometry (XRD) patterns of Artemisia sphaerocephala Krasch. gum (ASKG) (**a**), anionic cellulose nanofiber (ACNF) (**b**), AFR0 (**c**), AFR5 (**d**), AFR10 (**e**), and AFR15 (**f**) films.

**Figure 7 polymers-12-00247-f007:**
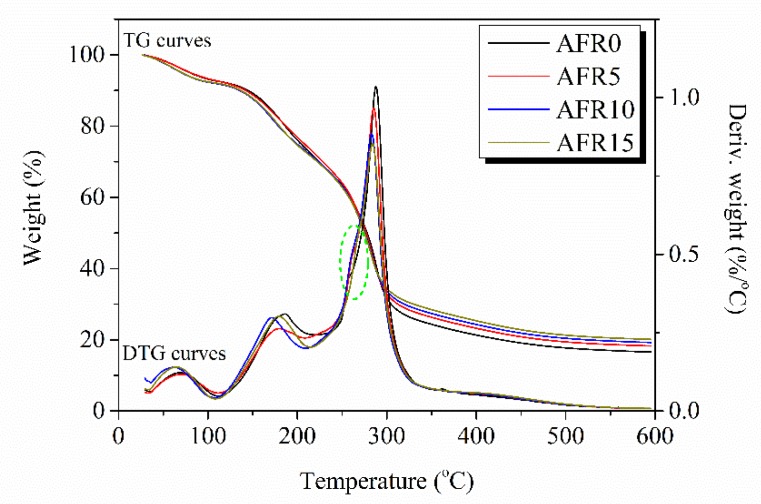
Thermogravimetric (TG) and derivative thermogravimetric (DTG) curves of ASKG/ACNF/RCE intelligent films.

**Figure 8 polymers-12-00247-f008:**
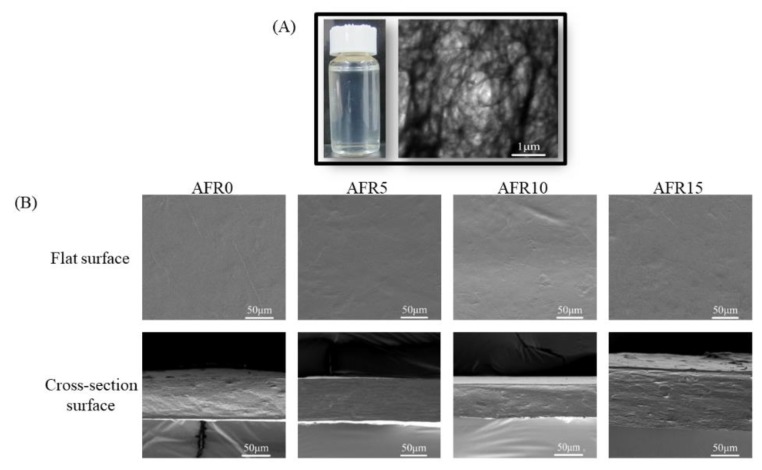
TEM picture (5000 magnification) of ACNF (**A**), SEM pictures (500 magnification) of the flat and cross-section surface morphology of intelligent films (**B**).

**Figure 9 polymers-12-00247-f009:**
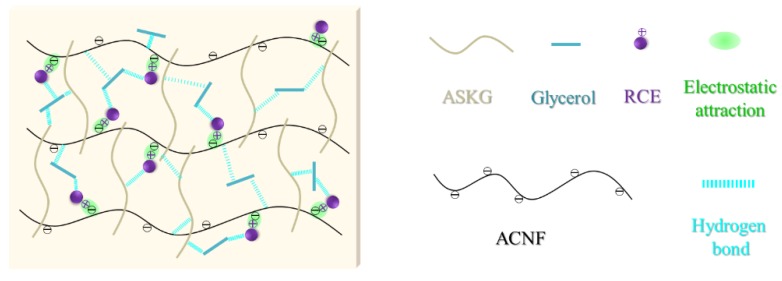
Effect of the RCE amount on mechanical properties of intelligent films.

**Figure 10 polymers-12-00247-f010:**
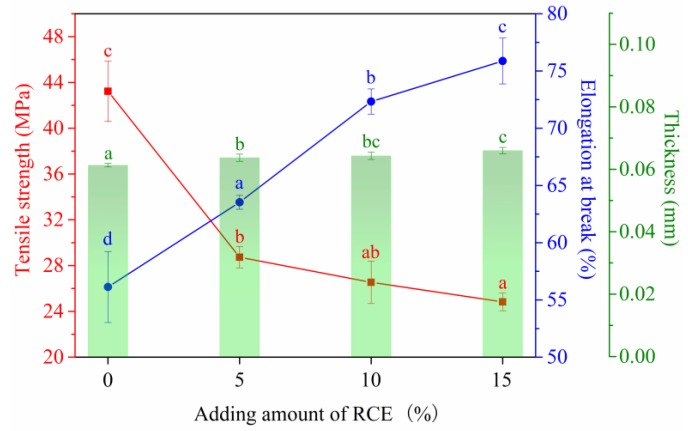
Effect of the RCE amount on mechanical properties of intelligent films.

**Figure 11 polymers-12-00247-f011:**
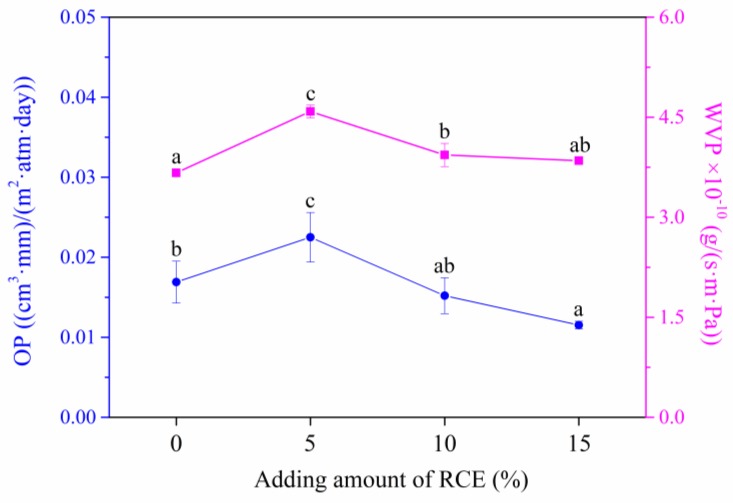
Effect of the RCE amount on oxygen barrier and water vapor barrier properties of intelligent films.

**Figure 12 polymers-12-00247-f012:**
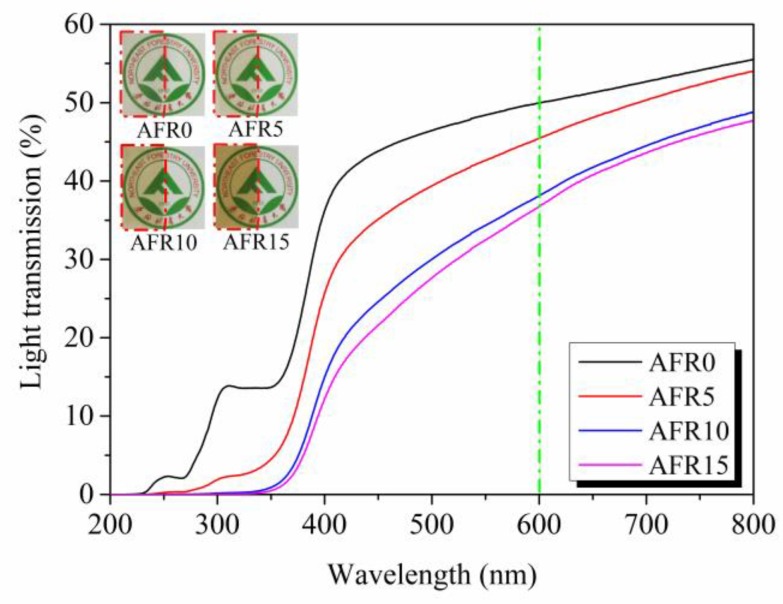
Effect of the RCE amount on light transparency properties of intelligent films.

**Figure 13 polymers-12-00247-f013:**
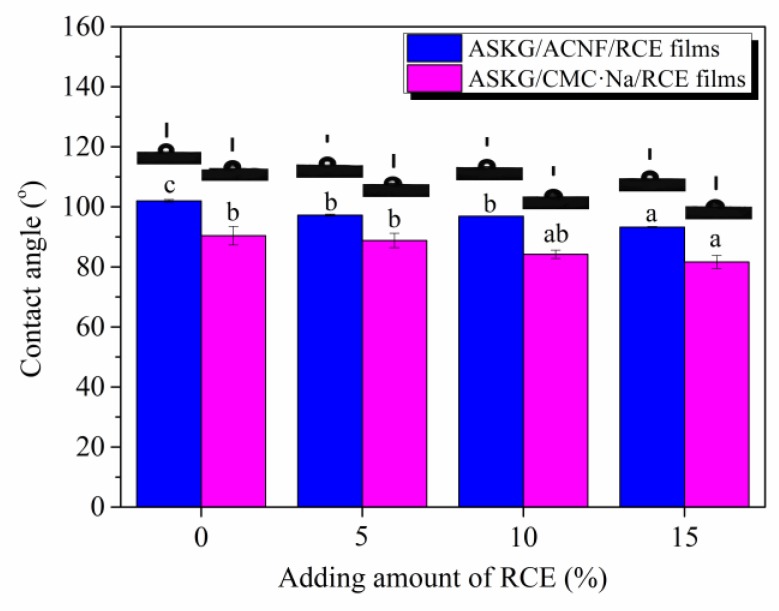
Effect of the RCE amount on water contact angle of ASKG/ACNF/RCE intelligent films and ASKG/CMC·Na/RCE intelligent films.

**Figure 14 polymers-12-00247-f014:**
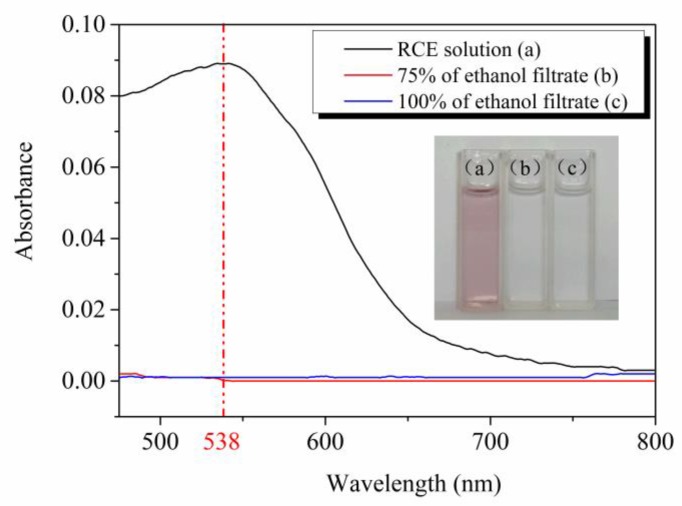
Visible light absorption curves of RCE solution and filtrates.

**Figure 15 polymers-12-00247-f015:**
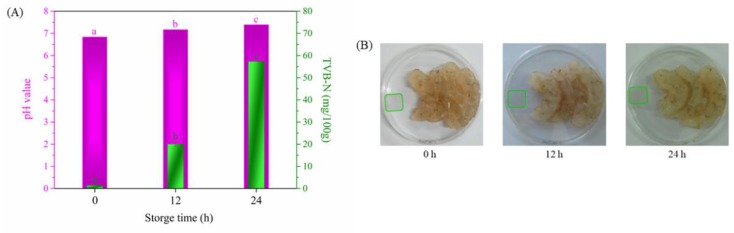
The pH and total volatile basic nitrogen (TVB-N) values of freshness shrimp, colorimetric parameters (**A**) and photograph (**B**) of intelligent at different storage times.

**Table 1 polymers-12-00247-t001:** The fitting data of Cross model.

Film-Forming Solution	*η*_0_ (Pa·s)	*K* (s)	*p*	*R* ^2^
AFR0	1.3857 ± 0.8004	2.1880 ± 0.1963	0.7639 ± 0.0315	0.9994
AFR5	1.3538 ± 0.3542	2.1631 ± 0.0974	0.7236 ± 0.0138	0.9999
AFR10	1.3250 ± 0.4616	1.9931 ± 0.1225	0.7348 ± 0.0206	0.9997
AFR15	1.3015 ± 0.3603	1.8871 ± 0.0955	0.7350 ± 0.0175	0.9998

**Table 2 polymers-12-00247-t002:** The colorimetric parameters and photographs of intelligent films under different buffer solutions.

Sample	pH	*L**	*a**	*b**	Δ*E*	Before	After
AFR5	3.0	79.91 ± 0.36 ^a^	6.13 ± 0.40 ^f^	−2.98 ± 0.39 ^a^	6.18 ± 0.31 ^d^	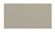	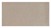
4.0	81.31 ± 0.75 ^b^	2.86 ± 0.08 ^e^	−3.47 ± 0.86 ^a^	4.84 ± 0.40 ^cd^	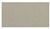	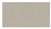
5.0	82.01 ± 0.47 ^bc^	2.24 ± 0.03 ^d^	−3.80 ± 0.73 ^a^	4.92 ± 1.38 ^cd^	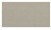	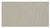
6.0	81.81 ± 0.43 ^b^	2.02 ± 0.02 ^d^	−3.31 ± 0.73 ^a^	4.23 ± 0.23 ^bc^	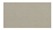	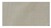
7.0	81.00 ± 0.86 ^ab^	1.24 ± 0.03 ^c^	−1.19 ± 1.37 ^b^	2.82 ± 1.70 ^ab^	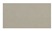	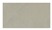
8.0	83.12 ± 0.36 ^c^	1.33 ± 0.05 ^c^	−4.43 ± 0.35 ^a^	4.61 ± 0.55 ^cd^	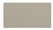	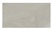
9.0	82.05 ± 0.58 ^bc^	−0.48 ± 0.30 ^b^	−0.87 ± 1.12 ^c^	2.37 ± 0.53 ^a^	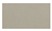	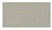
10.0	79.90 ± 0.97 ^a^	−4.08 ± 0.39 ^a^	−0.72 ± 1.09 ^c^	6.12 ± 0.37 ^d^	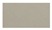	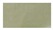
AFR10	3.0	78.71 ± 0.20 ^b^	5.86 ± 0.33 ^f^	0.48 ± 0.49 ^b^	10.46 ± 1.02 ^bc^	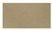	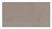
4.0	78.08 ± 1.01 ^b^	3.15 ± 0.04 ^e^	2.62 ± 1.04 ^bc^	8.59 ± 0.62 ^abc^	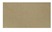	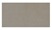
5.0	77.92 ± 0.22 ^b^	2.69 ± 0.18 ^de^	3.55 ± 0.58 ^cd^	8.871 ± 0.77 ^abc^	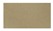	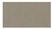
6.0	77.20 ± 0.37 ^b^	2.44 ± 0.07 ^cde^	3.94 ± 0.24 ^cd^	8.54 ± 0.51 ^abc^	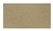	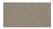
7.0	77.89 ± 1.52 ^b^	1.93 ± 0.20 ^bc^	4.25 ± 2.86 ^cd^	7.64 ± 3.35 ^a^	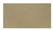	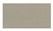
8.0	81.31 ± 0.77 ^c^	1.98 ± 0.06 ^bcd^	−2.39 ± 1.14 ^a^	11.22 ± 1.55 ^c^	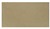	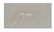
9.0	77.48 ± 0.60 ^b^	1.66 ± 0.22 ^b^	4.50 ± 0.37 ^cd^	7.10 ± 0.21 ^a^	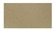	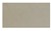
10.0	73.76 ± 0.76 ^a^	−2.83 ± 1.01 ^a^	5.36 ± 1.94 ^d^	8.16 ± 0.68 ^ab^	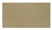	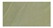
AFR15	3.0	77.82 ± 0.77 ^c^	6.36 ± 0.46 ^e^	3.24 ± 0.97^b^	12.48 ± 0.83 ^ab^	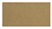	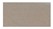
4.0	78.27 ± 2.53 ^cd^	3.93 ± 0.78 ^d^	4.08 ± 4.68 ^b^	12.14 ± 4.49 ^ab^	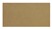	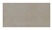
5.0	77.51 ± 0.76 ^bc^	3.08 ± 0.27 ^cd^	6.33 ± 1.04 ^bc^	10.37 ± 1.28 ^ab^	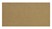	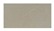
6.0	79.20 ± 0.31 ^cd^	2.57 ± 0.08 ^bc^	2.83 ± 0.54 ^b^	14.05 ± 1.03 ^bc^	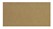	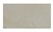
7.0	75.48 ± 0.85 ^b^	2.51 ± 0.14 ^bc^	8.94 ± 1.10 ^c^	10.14 ± 1.08 ^a^	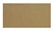	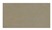
8.0	80.38 ± 0.10 ^d^	2.25 ± 0.01 ^bc^	−0.92 ± 0.61 ^a^	16.53 ± 0.29 ^c^	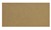	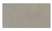
9.0	78.05 ± 1.04 ^c^	1.80 ± 0.19 ^b^	3.70 ± 1.78 ^b^	12.02 ± 2.33 ^ab^	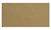	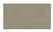
10.0	72.55 ± 1.28 ^a^	−1.95 ± 1.09 ^a^	5.83 ± 2.33 ^bc^	11.45 ± 1.34 ^ab^	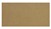	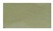

^a–f^ are the significant differences in the same parameters (*p* < 0.05).

**Table 3 polymers-12-00247-t003:** The colorimetric parameters and photographs of intelligent films under different humidities.

Sample	Humidity	*L**	*a**	*b**	Δ*E*	Before	After
AFR5	33%	77.73 ±0.39 ^a^	−8.47 ± 0.35 ^a^	17.57 ± 1.08 ^c^	20.29 ± 1.47 ^c^	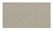	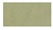
75%	78.92 ± 0.08 ^b^	−6.09 ± 0.12 ^b^	11.87 ± 0.22 ^b^	14.82 ± 0.19 ^b^	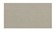	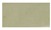
90%	79.34 ± 0.25 ^b^	−0.11 ± 0.07 ^c^	2.67 ± 0.32 ^a^	4.66 ± 0.18 ^a^	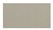	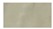
AFR10	33%	70.66 ± 0.44 ^a^	−7.22 ± 0.03 ^a^	31.95 ± 0.81 ^c^	25.02 ± 1.10 ^c^	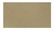	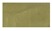
75%	72.21 ± 0.38 ^b^	−5.51 ± 0.11 ^b^	25.41 ± 1.19 ^b^	19.22 ± 1.69 ^b^	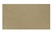	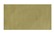
90%	73.97 ± 0.55 ^c^	−2.25 ± 0.38 ^c^	15.32 ± 1.24 ^a^	10.89 ± 1.10 ^a^	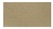	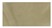
AFR15	33%	69.45 ± 0.31 ^a^	−4.92 ± 0.22 ^a^	31.48 ± 0.73 ^c^	20.96 ± 1.00 ^c^	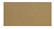	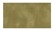
75%	69.48 ± 0.32 ^a^	−3.51 ± 0.28 ^b^	28.20 ± 0.64 ^b^	17.09 ± 0.75 ^b^	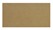	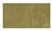
90%	68.73 ± 0.47 ^a^	−0.70 ± 0.11 ^c^	24.21 ± 0.93 ^a^	12.29 ± 0.51 ^a^	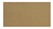	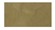

^a–c^ are the significant differences in the same parameters (*p* < 0.05).

**Table 4 polymers-12-00247-t004:** The colorimetric parameters and photographs of intelligent film.

Storage Time (h)	*L**	*a**	*b**	Δ*E*	Photograph
0	75.09 ± 0.58 ^ab^	0.32 ± 0.03 ^c^	−0.31 ± 0.52 ^a^	—	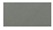
12	75.78 ± 0.99 ^b^	−4.18 ± 0.19 ^b^	7.61 ± 0.12 ^b^	9.22 ± 0.05 ^b^	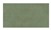
24	73.75 ± 0.65 ^a^	−5.05 ± 0.21 ^a^	10.35 ± 0.41 ^c^	12.04 ± 0.83 ^c^	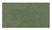

^a–c^ are the significant differences in the same parameters (*p* < 0.05).

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
