# Peer review of "Preparation, Characterization and Application of a Low Water-Sensitive Artemisia sphaerocephala Krasch. Gum Intelligent Film Incorporated with Anionic Cellulose Nanofiber as a Reinforcing Component"

_polymers, 2020, doi:10.3390/polym12010247_

Round 1

Reviewer 1 Report

In this paper, an efficient and versatile film (ASKG/ACNF/RCE intelligent film) is proposed. In general, the experiments were quite well designed and the authors provided characteristic results of the materials from the zeta potential, rheology, and Fourier-transform infrared spectroscopy, X-ray diffractometry and release results. It can be used in the field of intelligent packaging and gas sensor label. In my views, this paper can be accepted after minor revision. Below are some points that I would like to bring to the authors' attention: 1.Please supplement the extraction source and preparation process of “ RCE ”. 2. Provide references on the preparation of ASKG/ACNF/RCE intelligent film. 3. It is hoped that the author will consider the water sensitive effect of the ASKG film without ACNF. 4. Add the potential value of RCE and ACNF in potential measurement. 5. Mark the characteristic peak of infrared image. 6. Considering the economic effect, whether the film prepared by the author can be recycled? 7. In the introduction, the author said that naturally degradable and edible intelligent films or labels have received extensive attention from researchers. If the addition of ACNF weakens its water sensitivity, will it increase the difficulty of degradation.

Reviewer 2 Report

The article represents a hydrophobic film having natural indicator uses for packaging applications. The motivation of the article is well enough however the novelty of this manuscript needs to be addressed again with comparing with commercially available materials.

Apart from that some technical and methodological queries to be explained before its final acceptance:

Clear the motivation of choosing each material? Say, if hydrophobic material is uttermost important then why authors used anionic cellulosic form? Is CNC not properly dispersed in aqueous solution? Please add all the experimental details in supporting information that readers can understand the technique and methods that have been followed.  Abstract and conclusion sections need to be cultivated with the point-to-point hypothesis and experimental results. Clear the mechanism behind the preparation of the film. Why the as-synthesized film is supposed to be useful in packaging applications? Add at least one picture of the film. 

Reviewer 3 Report

Authors claim that „Red cabbage extracts (RCE) was selected as a sensitive indicator” This statement requires further explanation since it has not been stated what RCE is the indicator of.

Authors state that “The developed intelligent films could respond to buffer solutions and NH3”  This claim also requires further explanation and justification.

Line 42 – “according to recent reports” has been used twice.

In the section devoted to the ”Preparation of ASKG/ACNF/RCE intelligent films”: The volume of water, the size and type of material used for casting method and the volume of casting mixture have to be included. Moreover, the thickness of the obtained films has to be indicated.

During the Release test different concentrations of ethanol solution were used. The values of ethanol concentration have to be indicated.

In the case of hydrophilic films the measurement of contact angle of water has not been performed properly. It is well known that water often diffuses into analyzed material therefore, measurements are best carried out using bipolar glycerin. For this reason I suggest performing the contact angle analysis again, by applying glycerin for this procedure.

There is no indication as to the type of buffer solutions used during different analyses.

I do not see any differences in FTIR spectra (figure 5) after the addition of RCE. The Authors have to present FTIR results in a more legible way, for example by arranging all of the spectra together. The same conclusion can be drawn in relation to the XRD data  (figure 6) No changes are visible after the addition of RCE. In my opinion the discussion of the obtained results is not sufficiently supported by the figures in either of the cases.

In figure 7 the thermograms of particular components of the obtained materials have to be shown.

In aim to analyze the surface of the obtained materials the AFM technique has to be applied.

Summarizing, in my opinion:

The results of FTIR, TG and XRD are not consistent with the conclusions.

XRD, as well as TG analyses, can be removed while AFM and contact angle measurements need to be included.

The discussion of the obtained results is rather inadequate and needs to be improved.

I suggest a major revision of the manuscript submitted for review.

Reviewer 4 Report

The paper entitled “Preparation, characterization and application of a low water-sensitive Artemisia sphaerocephala Krasch. gum intelligent film incorporated with anionic cellulose nanofiber as a reinforcing component” by Liang T. and Wang L. has been reviewed. The paper deals with the elaboration of intelligent films with red cabbage extracts as a sensitive indicator. The experimental results seem to be very promising. But there are some remarks which should be taken into consideration before publication. To begin with, there are some mistakes in the English language (for example, line 42; line 66; line 203; line 235, and so on).

1.       Line 17: barrier properties towards what?

2.       Line 87: are you sure that all water was removed under these conditions?

3.       Line 99: the value of “room temperature” should be given.

4.       Lines 118-121: at which atmosphere the thermogravimetrical analysis was performed?

5.       Lines 202-211: the details of Cross model should be given in the text.

6.       Section 3.3: the values of FTIR bands should be added to Fig. 5.

7.       Lines 233-238: the authors noted about the enhancement of the band. Were the spectra normalized?

8.       Section 3.4: the values of XRD peaks should be added to Fig. 6. Besides, are the authors sure about the precision (two digits it is too precise)?

9.       Line 248: the authors noted the changes of the crystal structure of the intelligent film. The structure should be proposed.

10.   Lines 252-256: the temperature of the second weight loss (at 186.04 °C (also too precise)) attributed to glycerol decomposition is different as a function of the RCE content. This fact should be explained.

11.   Figure 8: the difference between images should be given in Figure Caption.

12.   Section 3.7: the complete names for TS and EB should be given.

13.   Lines 280: the authors stated that “OP and WVP of the intelligent film that contained RCE were higher”. This sentence should be rephrased for clarity. For example, the RCE content should be given.

14.   Lines 283-286 and line 370: the authors explained the observed fil properties by the “new tight network structure”. This structure should be proposed. In addition, what does it mean “the developed intelligent films remained excellent”? Excellent in terms of what?

Reviewer 5 Report

The manuscript presented interesting results concerning the film sensibility to pH change, which might allow its use as intelligent film for food packaging. The introduction is well-written.

On the other hand, in general, experimental results are under evaluated and discussion is quite superficial/speculative:

Tittle:

Preparation, characterization and application of a low water-sensitive Artemisia sphaerocephala Krasch.  gum intelligent film incorporated with anionic cellulose nanofiber as a reinforcing component

What was reinforced by incorporation of anionic cellulose nanofiber?

Line 187-190

The related data of AFR0, AFR5, AFR10 and AFR15 solutions in Fig. 2 indicate that the charge of the AFR0 solution is –17.4 mV, which is related to the charge of COO- in ACNF [38]. When RCE   increased from 0% to 15%, it also increased from –17.4 to –12.3 mV, which indicates that an electrostatic interaction between the flavylium cations in RCE and the COO- in ACNF occurred.

 Indeed, the incorporation of RCE decreased the Zeta Potential.  However authors cannot affirm that such decreasing is consequence of electrostatic interaction between the flavylium cations in RCE and the COO- in ACNF. It is speculative, it was based on an unique reported paper.  Relationship between RCE and Zeta Potential decreasing must be demonstrated.

Line 233-235

After RCE addition, the band at 3289 cm−1 shifted to a low wavenumber and the band at 2924 cm−1 was enhanced, which indicates that the hydrogen bonds among the ASKG, ACNF and glycerol were broken and new hydrogen bonds formed.

It lacks of sense! Authors add RCE to the system and claim that shift of OH band is consequence of hydrogen bondong break and new formation? There are many other feasible reasons for the band shift, how to be sure about hydrogen bonding changes? Were the spectra normalized? Beside on what theory, the increase of the peak located at 2924 cm-1 can be assigned to hydrogen bonding changes? Peaks must be assigned in the spectra.

Line 274-278

Figure 9 shows that the TS of the intelligent films decreased from 43.23 to 24.83 MPa but the EB of the intelligent films increased from 56.13% to 75.87% with increasing RCE. These results are assigned to the electrostatic attraction between RCE and ACNF and the formation of new hydrogen  bonds among film-forming contents, which altered the hydrogen bond arrangement and promoted molecular chain motion in the intelligent films [49].

Once again, authors tried to explain molecular interactions using macroscopic experiments. It does not make sense.  Discussion must be rewritten.

 Lines 297-307

The water-contact-angle data of the ASKG/ACNF/RCE and ASKG/CMC·Na/RCE intelligent  films are shown in Fig. 12. It shows that the contact angle of the AFR0 film was 102°, which indicates a low water sensitivity, owing to the excessively tight network structure of the AFR0 film and the hydrophobic properties of the ACNF.

 (Why? Please, demonstrate!)

An increase of RCE resulted in a decrease of contact angle. However, the contact angle remained at 93.25° when the amount of RCE added reached 15%, which indicates a low water sensitivity. These results were attributed to RCE addition that broke the tight structure that formed among the ASKG, ACNF and glycerol, and promoted water penetration.

 (Why? Based on what can authors affirm it?) Contact angle measurements cannot be applied for water absorbing materials!

Compared with the ASKG/CMC·Na/RCE intelligent films in our previous work [25], the contact angle of films that contained ACNF were higher than those that contained CMC·Na, which made them suitable for use under higher humidity conditions.

(Why? Following what previous reported parameters or standard? )

The above discussions (and further conclusions) must be improved before publication.

Round 2

Reviewer 3 Report

The manuscript has been significantly improved. All indicated discrepancies have been corrected.

I suggest acceptance of the submitted manuscript.

Reviewer 5 Report

Authors refused to modify the manuscript in many points. They prefered to justify the superficial and especulative discussion using previously published reports, pushing forward some mistakes.